# Demonstration of dual Shapiro steps in small Josephson junctions

Fabian Kaap ⬤ ✉, Christoph Kissling ⬤ , Victor Gaydamachenko, Lukas Grünhaupt ⬤ & Sergey Lotkhov ⬤

Bloch oscillations in small Josephson junctions were predicted theoretically as the quantum dual to Josephson oscillations. A significant consequence of this prediction is the emergence of quantized current steps, so-called dual Shapiro steps, when synchronizing Bloch oscillations to an external microwave signal. These steps potentially enable a fundamental standard of current $I$, defined via the frequency $f$ of the external signal and the elementary charge $e$, $I = \pm n \times 2ef$, where $n$ is a natural number. Here, we realize this fundamental relation by synchronizing the Bloch oscillations in small Al/AlO$_x$/Al Josephson junctions to sinusoidal drives with frequencies from 1 to 6 GHz and observe dual Shapiro steps up to $I \approx 3$ nA. Inspired by today's voltage standards and to further confirm the duality relation, we investigate a pulsed drive regime and observe an asymmetric pattern of dual Shapiro steps. This work confirms quantum duality effects in Josephson junctions and paves the way towards a range of applications in quantum metrology based on well-established fabrication techniques and straightforward circuit design.

Today's voltage standard provided by national metrology institutes[1] relies on the synchronization of Josephson oscillations[2] with an external rf drive. This leads to equidistant voltage steps, the so-called Shapiro steps[3]. The dual effect of quantized current steps was predicted theoretically almost 40 years ago[4,5] and originates from the quantum mechanical description of small Josephson junctions, where charge and phase are conjugate variables. Quantized currents based on this duality could potentially close the metrological triangle, uniting the ampere, the volt and the ohm through fundamental constants and frequency. The closure of the triangle would open the way for combining macroscopic quantum relations of voltage, current, and resistance in one device and allow to investigate the consistency of the three relations. In terms of implementing a quantum current standard, significant advantages of using Josephson junctions for this purpose, as compared to e.g. semiconductor pumps[6,7], are higher frequencies in the GHz range and technological compatibility with Josephson voltage standards.

To observe dual Shapiro steps, the Josephson junction has to be embedded in a high impedance environment $Z_{env} > R_Q = h/4e^2 \approx 6.4$ kΩ in order to suppress quantum fluctuations of charge and enable the observation of charging effects. The first indications of dual Shapiro steps were reported a few years following the original prediction[8] and relied on a purely resistive environment. However, strong heating effects made it difficult to observe quantized current steps[9], such that no detailed study of the synchronization mechanism could be carried out. To solve this issue a biasing circuit combining high-ohmic off-chip resistors and compact on-chip inductances[10], whose specific impedance exceeds the quantum resistance was proposed. Such so-called superinductances can in principle be realized via Josephson junction chains[11–14], nanowires on membranes[15,16] or high kinetic inductance leads[17–24] of typically a few tens of $\mu$m length. Such elements together with small Josephson junctions allow to use capacitive coupling for synchronization of Bloch oscillations (BO) and thus, mitigate rf-heating in the resistors.

Here, we utilize the large resistivity of oxidized titanium[25] and the kinetic inductance of granular aluminum[19,26] to implement the high impedance environment and combine it with standard Al/AlO$_x$/Al Josephson junction fabrication technique[27] to realize an experiment guided by the theoretical proposals[4,5,10]. We demonstrate pronounced quantized current steps in small Josephson junctions by synchronizing

Physikalisch-Technische Bundesanstalt, Bundesallee 100, 38116 Braunschweig, Germany. ✉e-mail: fabian.kaap@ptb.de

their BO to external microwave signals between 1 and 6 GHz, without relying on discrete high-quality resonant modes, which limited recent previous evidence of dual Shapiro steps to four distinct frequencies[14]. Using shadow evaporated Josephson junctions, we provide an alternative to coherent quantum phase slip circuits, based on advanced fabrication techniques[22], which demonstrated the first pronounced quantized current steps overcoming the heating challenge of initial experiments[8,9]. To further investigate the duality relation, we implement pulsed external signals for synchronization of BO, a technique to enhance the width of quantized voltage steps[28,29]. We demonstrate an asymmetric pattern of dual Shapiro steps consistent with theoretical predictions[30], reinforcing the observed effect as the dual to quantized voltage steps.

The behavior of a Josephson junction is determined by its charging energy $E_C = \frac{e^2}{2C}$ and its Josephson energy $E_J = \frac{\Phi_0 I_c}{2\pi}$, where $e$, $\Phi_0$ are the electron charge and the flux quantum and $C$, $I_c$ are the capacitance and critical current of the Josephson junction. Taking into account that the charge and phase operators are governed by the canonical commutation relation $[\hat{Q}, \hat{\varphi}] = 2ei$, the Hamiltonian is given by

$$\hat{H} = \frac{\hat{Q}^2}{2C} - E_J \cos(\hat{\varphi}). \qquad (1)$$

This Hamiltonian can be solved using Blochs theorem[31,32] and yields periodic energy bands $E_s(q) = E_s(q + 2e)$, where $s \in \mathbb{N}_0$ is the level number and $q$ is the so-called quasicharge, which represents the charge injected to the junction[4,5].

In Fig. 1a the two lowest energy bands ($s = 0, 1$) are shown for different ratios $E_C/E_J$. In the ground-state approximation, the dynamics of this system is given by the so-called Langevin equation:

$$\dot{q} = I_{bias} - V[q(t)]/R + I_{rf}(t) + \tilde{I}(t), \qquad (2)$$

with $I_{bias}$ a dc bias current, $V = \frac{\partial E_0}{\partial q}$, $R$ the real part of the impedance of the environment, which we for simplicity assume to be purely ohmic, $\tilde{I}(t)$ the fluctuation current and $I_{rf}$ an external rf-signal[4,5]. If $I_{rf}(t) = 0$, $\tilde{I}(t) = 0$ and $I_{bias} < \max[\frac{\partial E_0}{\partial q}]/R$, there is a stationary solution for Eq. (2), a Coulomb blockade. For currents $I_{bias} > \max[\frac{\partial E_0}{\partial q}]/R$ the solution is periodic in time and results in BO with frequency $f \approx I_{bias}/2e$ [see Fig. 1b]. These BO can synchronize to an external rf current $I_{rf}$,

generating quantized current steps at $I = \pm n \times 2ef$, referred to as dual Shapiro steps, where $n$ is a natural number.

Figure 1c–e shows optical and scanning electron microscope (SEM) pictures of a representative device. We fabricate the chip using three lithographical steps on a thermally oxidized silicon wafer[24]. First, we pattern a wiring structure from Au (not shown) using optical lift-off lithography. Subsequently, we fabricate 150 nm wide and 15 nm thick TiO$_x$ resistors[25]. Finally, we in situ combine a 20 nm thick granular aluminum superinductor[19] with double-angle evaporation Al/AlO$_x$/Al Josephson junctions[27]. To implement the last lithographic step we utilize a PMMA/copolymer liftoff mask with a controllable undercut[33]. Each of the two Josephson junctions of the SQUID has an area $A_{JJ} \approx 20 \times 30$ nm [see Fig. 1e] leading to an upper bound of the charging energy $E_C = e^2/(4A_{JJ} \times 50\ \text{fF}\mu\text{m}^{-2}) = 1.3$ meV. The true experimental value of $E_C$, however, includes the full 3D junction topology as well as the stray capacitances of the leads, and is measured to be $E_C \approx 80\ \mu$eV (see Supplementary Fig. 3). In order to keep the Josephson energy $E_J$ comparable to the charging energy, we dynamically oxidize at $P_{O_2} = 0.1$Pa for 300s, resulting in $E_J \approx 60\mu$eV (see Supplementary Fig. 3). The leads of the SQUID are connected to granular aluminum strips, which split up in two oxidized titanium resistors[25,34] [see Fig. 1c]. From a measurement of the leads with the cryostat at base temperature, we extract the resistance of the TiO$_x$ strips $R_{TiO_x} = 275$ kΩ. Using the Matthis–Bardeen formula we estimate $L_{grAl} = 0.18\hbar R/(k_B T_c) \approx 50$ nH, from a room temperature resistance measurement of the grAl strip. Aside from their inductive contribution leading to an characteristic impedance $Z_{grAl} > R_Q$, the grAl strips might act as quasiparticle filters due to their larger superconducting energy gap as compared to the Al films[35]. Quasiparticle tunneling is a process that partially replaces the coherent Cooper pair tunneling associated with Bloch oscillations. Reducing the presence of quasiparticles in the vicinity of the Josephson junction will thus result in a larger Coulomb blockade. An external rf-signal for synchronization of the BO is supplied via a 50 Ω slot line and capacitively couples to the SQUID [see Fig. 1c, d].

## Results

All measurements were performed in a dilution refrigerator at ~15mK. All dc lines include three meter long Thermocoax™ cable filters[36] connecting the room temperature electronics to the mK sample stage. The rf driving signals applied to the sample are attenuated by 50 dB at

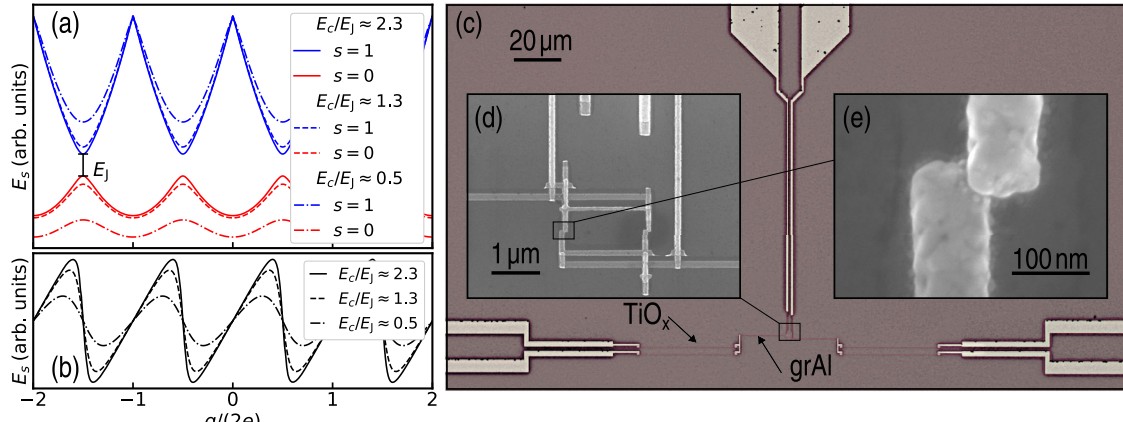

**Fig. 1 | Bloch oscillations: energy diagram and experimental realization. a** Energy $E_s(q)$ of the two lowest Bloch bands for different ratios $E_C/E_J$. **b** Ground state oscillations of voltage calculated by taking the derivative $\frac{\partial E_0}{\partial q} = V$. These so-called Bloch oscillations have a frequency $f_B \approx I_{bias}/2e$ (see main text). **c–e** Optical and scanning electron microscope images of a representative device consisting of a dc-SQUID [see panel (**d**)] with two Josephson junctions of area $A_{JJ} \approx 20 \times 30$ nm [see panel (**e**)] embedded in a bias circuitry. The slot line from the top delivers the

rf-driving signal via a capacitive coupling with $C_c \approx 0.3$ fF while the horizontal leads are used for four-probe dc-measurements. The bias circuitry includes the 20 $\mu$m long granular aluminum inductor[19] with kinetic inductance $L_{grAl} \approx 50$ nH and characteristic impedance $Z_{grAl} \approx 10.5$ kΩ > $R_Q = h/4e^2 \approx 6.4$ kΩ and the 40 $\mu$m long TiO$_x$ resistors[25,34] of $R_{TiO_x} \approx 275$ kΩ. This provides a high impedance environment $Z_{env} \geq R_Q$, suppresses quantum fluctuations of charge and enables Bloch oscillations[4].

different temperature stages of the dilution cryostat (see Supplementary Fig. 1). The rf-powers quoted in the following refer to the power level at the mK-stage, where additional cable losses are neglected and should be equal for all experiments.

## Coulomb blockade

Figure 2a shows the *IV* curves of the sample without external rf-drive for values of the external flux between $0 \leq \Phi/\Phi_0 \leq 0.45$. As can be seen, the effective Josephson energy of the dc-SQUID $E_J(\Phi) = E_{J,0}|\cos(\pi\frac{\Phi}{\Phi_0})|$, therefore the shape of the Bloch bands and, consequently, the *IV* curve are tuned. Close to $\Phi \approx \Phi_0/2$ the Coulomb blockade increases, which can be explained by the increasing amplitude of the first Bloch band [cf. Fig. 1a]. Moreover, for large bias currents $I_{bias}$ the probability to tunnel from $E_0$ to $E_1$ increases and the so-called Landau–Zener tunneling (LZT)[37–39] destroys the coherence of the BO. The LZT manifests in a resistive branch in the *IV*-characteristics as a positive slope at currents exceeding an LZT onset current of a few nA. Since the gap between the lowest two bands is approximately proportional to $E_J$, the onset current for this LZT branch rapidly decreases with the external

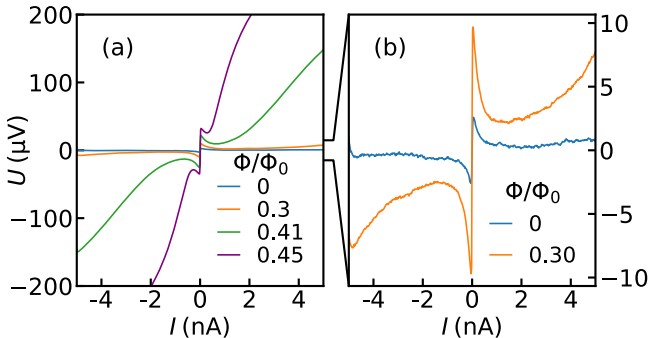

**Fig. 2 | External flux dependent Coulomb blockade of the device. a** *IV* curves of the sample [cf. Fig. 1c–e] for external flux values $0 \leq \Phi/\Phi_0 \leq 0.45$. By approaching $\Phi_0/2$ and thus increasing the ratio $E_C/E_J$ we observe wider ranges of the Coulomb blockade and steeper slopes of the resistive branch caused by Landau-Zener tunneling (LZT)[37–39] (see main text). **b** Zoom-in on the *IV* curves for $\Phi/\Phi_0 = 0$, 0.3. A back-bending to almost zero voltage can be observed, which is a hallmark of Bloch oscillations in a dc-measurement. We identify flux values $0 \leq \Phi/\Phi_0 \leq 0.3$ ($1.3 \leq E_C/E_J \leq 2.3$) as a suitable range for observing dual Shapiro steps (see Supplementary Fig. 7).

flux and reaches a minimum at $\Phi/\Phi_0 = 0.5$. From measurements at different cryostat temperatures, we estimate an effective electron temperature of 40mK (see Supplementary Fig. 6). Given this temperature $k_B T \ll \Delta^{(0)} = \min(E_1) - \max(E_0)$ the dynamics of the system are confined to the lowest energy band.

Figure 2b shows a zoom-in of the *IV* curves for $\Phi/\Phi_0 = 0$ and 0.3. In the blue curve a clear backbending from the initial Coulomb blockade of ~4 μV down to $U \approx 0$ can be observed, indicating negligible LZT up to currents of ~±5 nA. In comparison, the orange curve shows an extended blockade range of ~18 μV, but at the same time a reduced onset current of LZT of ~2nA. As a compromise between maximizing the Coulomb blockade width, which sets an upper limit for the dual Shapiro step width, and the range of quantized currents limited by LZT, all measurements shown in this work are done at $\Phi/\Phi_0 = 0.3$, resulting in $E_C/E_J \approx 2.3$ (see Supplementary Fig. 7).

## Dual Shapiro steps using sinusoidal drive

In the following we perform measurements with a sinusoidal drive applied to the SQUID at different GHz-frequencies and rf-power levels. Figure 3a shows the *IV* curves measured at a drive power of $p_{drive} = -49$ dBm and frequency of $f_{drive} = 2.8$GHz and 4.6GHz featuring two distinct steps around the quantized values of $I = \pm 2ef_{drive} \approx 0.93$ nA and 1.53 nA, respectively (brown and violet dotted lines). The Coulomb blockade as well as the dual Shapiro steps are skewed as compared to the non-rf case [see orange curve in Fig. 2b], which we attribute to additional heating caused by the external rf-drive. We estimate an electron temperature of ~250mK, which is a factor of ~6 higher than for the undriven measurements of the Coulomb blockade (see Supplementary Fig. 6). The continuous raise of voltage outside the range $|I| > 2$nA is caused by the LZT and resembles the LZT branch in the orange curve in Fig. 2b.

Figure 3b shows the differential resistance $R_{diff} = \frac{dU}{dI}$ measured with a lock-in amplifier for different drive frequencies $f_{drive}$ at a constant power $p_{drive} = -49$ dBm. The white dashed lines indicate the due positions of the first and second dual Shapiro step, $I_{peak} = \pm n \times 2ef_{drive}$. Up to frequencies of ~6GHz the first step can clearly be identified, while towards higher frequencies the steps are vanishing [see Fig. 3a] and the Coulomb blockade reappears, which indicates lower rf-power reaching the SQUID. We attribute this to stronger reflection of the rf-signal at higher frequencies in the slot line. Additionally, for synchronization at higher frequencies larger power is needed, since the width of (dual) Shapiro steps as a function of power scales proportional to

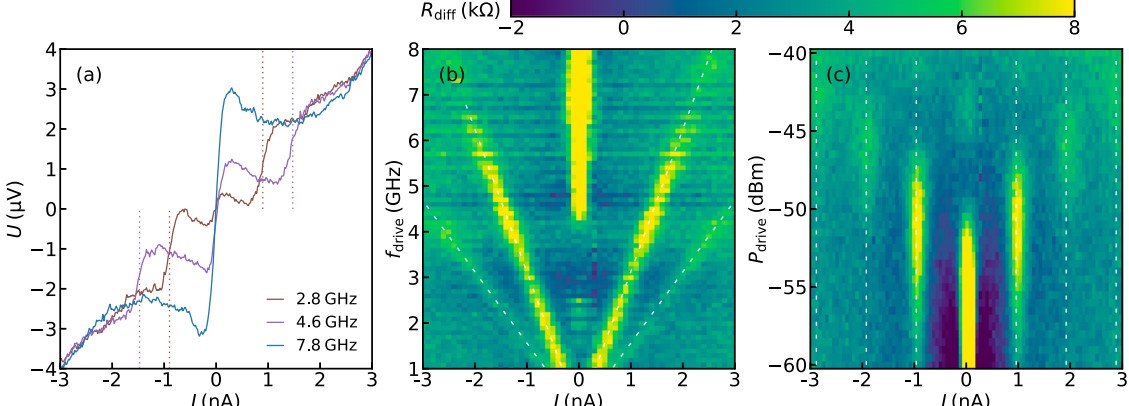

**Fig. 3 | Dual Shapiro steps due to sinusoidal rf-drive. a** Measured *IV* curve with $f_{drive} = 2.8$GHz(brown), 4.6GHz (violet), 7.8GHz(blue) for rf-power at the chip input of $p_{drive} = -49$ dBm. The dashed lines indicate the positions of the first dual Shapiro steps according to $I_{dSs} = \pm 2ef_{drive} \approx \pm 0.93$ nA and $\pm 1.53$nA. **b** Differential resistance $R_{diff} = \frac{dU}{dI}$ measured using a lock-in amplifier for different drive frequencies $f_{drive}$ at constant drive power $p_{drive} = -49$ dBm. Dashed white lines confirm the expected position of dual Shapiro steps according to $I_{dSs} = \pm 2ef_{drive}$. **c** $R_{diff}$ for different powers $p_{drive}$ at fixed frequency $f_{drive} = 3GHz$, leading to steps at $I_{dSs} \approx \pm n \cdot 0.96$nA. The amplitude of the Coulomb blockade and the dual Shapiro steps vary with $p_{drive}$ (see main text).

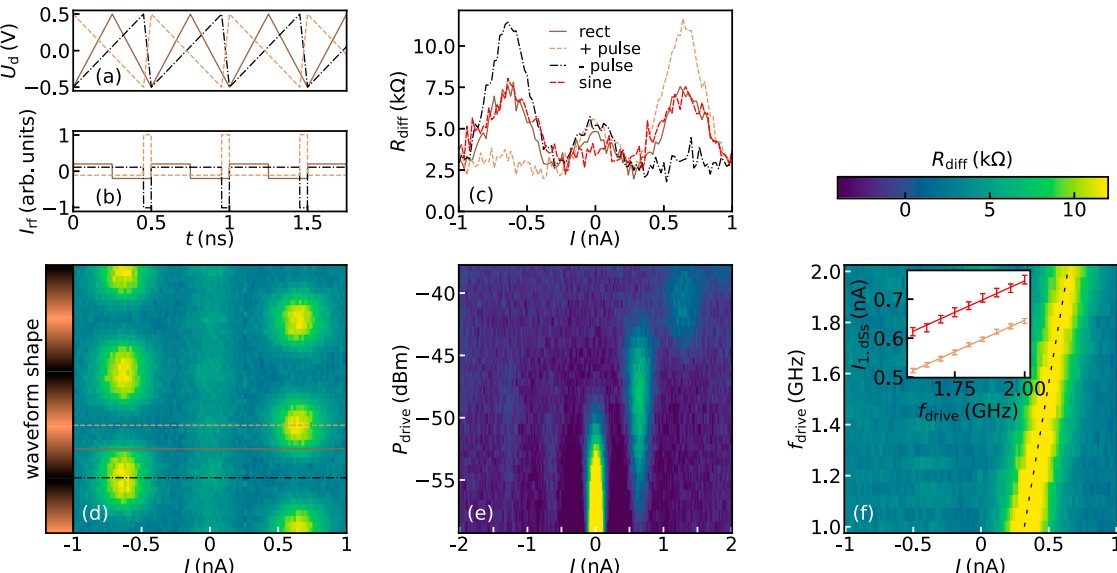

**Fig. 4 | Shaping dual Shapiro steps by using different waveforms. a** Dedicated waveforms $U_d(t)$ used to drive the dual Shapiro steps. **b** rf-current injection according to $I_{rf} \propto C_c \times \frac{dU_d}{dt}$. The symmetric waveform (brown) leads to a rectangular current profile, while the negative (positive) sloped sawtooth signal lead to short positive (negative) current pulses (black and beige, respectively).
**c** Differential resistance $R_{diff}$ for the three current profiles shown in (**b**) and a sinusoidal drive (red) with $f_{drive} = 2\,GHz$ and $p_{drive} = -48\,dBm$ (see Supplementary Fig. 4). Depending on the pulse, either the positive or negative dual Shapiro step is enhanced and reaches almost twice as high peak values $R_{diff}$ as compared to the symmetric drives (triangular and sinusoidal), while the opposite step is suppressed.
**d** $R_{diff}$ measured for linearly sweeping back and forth between a positive (beige) and negative pulsed drive (black). The colorbar on the y-axis indicates the gradual change of the sawtooth polarity. Colored lines mark the curves shown in (**c**).

**e** Power dependence of $R_{diff}$ using a pulsed drive at 2GHz. Increasing the power of the pulse leads to a gradual activation of the consecutive dual Shapiro steps.
**f** Differential resistance $R_{diff}$ for different frequencies $f_{drive}$ of the pulsed drive. Over the whole frequency range the position of the positive dual Shapiro step (black dashed line) aligns with $I_{dSs} = 2ef_{drive}$, while the negative step is suppressed. Inset: Measured current of the first dual Shapiro step as a function of the drive frequency for the sinusoidal drive (red and plotted with 0.1nA offset) and the pulsed drive (beige). From a linear fit we can extract the charge of an electron according to $I = 2ef_{drive}$. For the sinusoidal drive we extract $e_{sine} = (1.612 \pm 0.010) \times 10^{-19}\,C$ and for the pulsed drive $e_{pulse} = (1.610 \pm 0.005) \times 10^{-19}\,C$. The errorbars correspond to the standard deviation of the step position extracted by a least squares fit of a Gaussian (see Supplementary Fig. 4). Note that the errorbars for the pulsed drive are smaller.

$|J_n(\alpha)|^b$, where $J_n$ are the Bessel functions of the first kind, $\alpha \propto \sqrt{P_{drive}}/f_{drive}$ is a normalized power and $b$ is an exponent, which depends on the impedance of the environment[22,40].

Figure 3c shows the dependence of $R_{diff}$ for rf-power between $-60\,dBm \le p_{drive} \le -40\,dBm$ at a fixed drive frequency $f_{drive} = 3\,GHz$. The equidistant set of peaks in $R_{diff}(I)$ indicates the dual Shapiro steps separated by $\Delta I = 0.96\,nA$. At low power, $p_{drive} \approx -60\,dBm$, the Coulomb blockade peak dominates, while the first dual Shapiro steps are barely visible. By increasing $p_{drive}$ the first current steps grow larger, as indicated by the higher resistance peaks, and they reach the maximum peak height at $p_{drive} = -51\,dBm$, while the Coulomb blockade disappears. Up to the higher power levels around $p_{drive} \approx -47\,dBm$ the second, and around $-43\,dBm$ the third dual Shapiro steps can be seen. Higher-order dual Shapiro steps are not visible due to LZT destroying the coherence of the Bloch oscillations. This measured pattern of dual Shapiro steps can be modeled using Eq. (1) and qualitatively reproduces the data obtained in refs. 22,30,41 (see Supplementary Fig. 5). We note that for higher rf-power the increased heating leads to smearing of the features. As mentioned before, the power dependence follows the Bessel-like behavior. However, our data does not allow to extract the exact exponent $b$ of this dependence.

**Pulse-driven dual Shapiro steps**
To implement a pulsed drive for the SQUID we apply a sawtooth signal $U_d(t)$ to the slot line, which yields $I_{rf} \propto C_c \times dU/dt$. Figure 4a depicts the signal profile of three different waveforms, namely a positive sawtooth, a negative sawtooth and a symmetric triangular waveform. The derived rf-current through the SQUID is sketched in Fig. 4b. For the triangle we assume a symmetric rectangular current profile of low amplitude, while both sawtooth waveforms will generate a unipolar current pulse of larger amplitude. In Fig. 4c the differential resistance

$R_{diff}$ is shown for the three current profiles displayed in Fig. 4b. For comparison, we plot similar data obtained for an optimized sinusoidal drive with $f_{drive} = 2\,GHz$ and $p_{drive} = -48\,dBm$ (see Supplementary Fig. 4). For the triangular waveform both positive and negative branch exhibit a similar peak height as in the case with a sinusoidal drive. In contrast, if using a sawtooth-like waveform generating current pulses, the peak height of $R_{diff}$ of one of the two opposite steps is approximately doubled in magnitude, while the other step is suppressed (see Supplementary Fig. 5, cf. ref. 30). As can be seen in Fig. 4d, by changing the drive waveform between a positive and a negative sawtooth (pulses) [beige and black curves in Fig. 4a, b], we observe a modulation of measured curves between the three representative cases shown in Fig. 4c, which are indicated by the three colored lines in panel (d). We note, that the largest differential resistance $R_{diff,peak}$ of the measured steps occurs for maximally tilted sawtooth signals leading to the highest current pulse in Fig. 4b.

The power dependence of $R_{diff}$ measured with pulsed driving at 2GHz is shown in Fig. 4e. By increasing the power, the first dual Shapiro step appears at $-50\,dBm$, where the Coulomb blockade fades. Further increasing the power above $-48\,dBm$ decreases the first dual Shapiro step and the second step appears at $-43\,dBm$. This power dependence of the individual steps is in qualitative agreement with numerical simulations (see Supplementary Fig. 5, cf. ref. 30). Note that to apply drive powers $P_{drive} > -44\,dBm$, the total attenuation in the lines is reduced to 40 dB.

In Fig. 4f we plot $R_{diff}$ for pulsed drives at different frequencies up to 2GHz, showing that the asymmetry in the $IV$ curve persists and the position of the step follows the fundamental relation $I = 2ef_{drive}$ (black dashed line). To quantify the enhancement of the quantized current step due to the pulsed drive [see inset of Fig. 4f], we extract the position of the first dual Shapiro step and compare it to the case of a sinusoidal drive

(see Supplementary Fig. 4, shown with 0.1nA for visualization). The power of the sinusoidal drive, $p_{drive} = -48$ dBm, was set to maximize the step width and is comparable to the power of the sawtooth drives with −47.8 dBm. The extracted value of the electron charge for the pulsed drive $e_{pulse} = (1.610 \pm 0.005) \times 10^{-19}$ C is showing smaller statistical errors than for the sinusoidal case $e_{sine} = (1.612 \pm 0.010) \times 10^{-19}$ C.

## Discussion

In conclusion, we experimentally show the occurrence of Bloch oscillations in Josephson junctions embedded in a high impedance environment made from granular aluminum and oxidized titanium. By synchronizing the Bloch oscillations with an external sinusoidal rf-drive, we demonstrate pronounced dual Shapiro steps and quantized currents up to 3 nA. Using pulsed drives, we show an asymmetric pattern of the dual Shapiro steps. This research serves as a stepping stone for future metrological applications, namely the development of new quantum standards for currents in the nA-range and thus, for closing the metrological triangle. For future experiments, the detailed studies of dual Shapiro steps arising due to a wide variety of driving waveforms appear to be of particular interest. Combining the demonstrated technique of drives with further improvements in the circuit layout like an optimized rf-line geometry and using even smaller, point-like contacts could lead to even wider and more precise current steps. With those improvements, it might be possible to reach the current metrological precision of current standards realized via single electron pumps ($\approx$0.2 ppm[6]).

During the review of our manuscript we became aware of a preprint by Antonov et al.[42], demonstrating dual Shapiro steps in Al/AlO$_x$/Al Josephson junctions. However, we note that our result seem to contradict the observation of Antonov et al. that dual Shapiro steps are only observed in samples with a Coulomb blockade smaller than 5 $\mu$V.

## Method

### Device fabrication

The total device fabrication consists of three lithographic steps on a 380 $\mu$m thick silicon substrate with a 600nm thermal oxide layer. All metal layers are structured by lift-off. We thermally evaporate a 45nm thick gold layer on top of a 5nm titanium layer for improved adhesion. This first wiring structure is patterned by photolithography using a Süss MA6 contact aligner and contains bond pads to connect the chip to the sample holder. The two following lithography steps for the TiO$_x$ resistors and the Al structures utilize a commercial 100 kV electron beam writer. For these depositions we use a custom made electron beam evaporation system consisting of an evaporation chamber and a substrate chamber, in which the substrate can be tilted for shadow evaporation. To create the TiO$_x$ resistors we evaporate titanium at a rate of 2Å s$^{-1}$, while the oxygen pressure inside the substrate chamber is kept at $3e - 6$mbar. In order to enable good electrical contact with the following layers the ends of the TiO$_x$ resistors are sealed with a 25 nm AuPd layer deposited under an angle of 32°. For the granular aluminum film we evaporate Al under an angle of 0° at a rate of 2Ås$^{-1}$ while setting an oxygen pressure of $1.5e - 5$ mbar inside the substrate chamber. The following two Al films for the Josephson junctions are evaporated under angles of $\pm 17.5°$ at a rate of 3 Å s$^{-1}$ with a pressure of $5e - 8$mbar inside the substrate chamber.

### Measurement setup

The chip is placed in a copper holder box with an integrated superconducting coil to apply a magnetic field to change the magnetic flux threading the SQUID loop. This box is thermally anchored to the mixing chamber plate of a commercial dilution refrigerator with a base temperature of  -15mK. To provide sufficient shielding the sample box is enclosed by copper and $\mu$-metal shields, both anchored to the mixing chamber plate. The *IV* curves are taken with compact DAQ modules from National Instruments, while the measurements of the *IR*$_{diff}$ curves are measured with a Lock-in amplifier SR860 from Stanford Research Systems. The voltage drop across the SQUID is amplified by a factor of $10^4$ with a Femto DLPVA-100-F-D low-noise voltage amplifier.

## Data availability

The data that support the findings of this study are available on Zenodo under accession code https://doi.org/10.5281/zenodo.11919737.

## Code availability

The code to generate the numerical simulation in the supplementary information is available on Zenodo under accession code https://doi.org/10.5281/zenodo.11919737.

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

## Acknowledgements
The authors thankfully acknowledge useful discussions with A. B. Zorin, D. Scheer, F. Hassler, and M. Bieler. The authors also acknowledge technical support from H. Marx and N. Ubbelohde, and A. Fernández Scarioni, M. Schröder, J. Blohm, P. Hinze and T. Weimann. This work was supported by the Deutsche Forschungsgemeinschaft (DFG) under Grant No. LO 870/2-1 and under Germany's Excellence Strategy—EXC-2123 QuantumFrontiers—390837967.

## Author contributions
F.K. and S.L. designed the project and the sample. F.K. fabricated the sample and measured the data with input from L.G. and S.L. C.K. contributed to the design of the rf-components of the sample. V.G. and F.K. modeled the data. F.K., L.G. and S.L. wrote the manuscript with input from all authors.

## Funding

## Competing interests
The authors declare no competing interests.
