## [Peer Review File · Nature Communications]

Editorial note: Figure on Page 24 of the PRF is reproduced from 'Theory of the Bloch-wave oscillations in small Josephson junctions' Journal of Low Temperature Physics 59, 347 (1985).

REVIEWER COMMENTS

Reviewer #1 (Remarks to the Author):

In this work, the authors studied the dual Shapiro steps stemming from the synchronization of Bloch oscillations and an external rf-drive. The experiment was performed in Josephson junctions made of Al/AlO_x/Al, embedded in a high impedance environment. Quantized, but with a finite slope, current steps were observed. They further investigated the effect of pulsed drives of different forms, and reached a consistent behavior, especially the asymmetric pattern. However, at the current stage, I could not make a decision on the recommendation of publication since I have some serious concerns.

1. I have serious concerns on the data. Without the rf-drive, there is back-bending in the I-V curve, e.g., in Fig. 2. When the rf-drive is applied, the back-bending is reformed and the current steps appear, e.g., in Fig. 3. However, the back-bending does not disappear completely and is still there. Therefore, the differential resistance should have negative values. However, the minimum R_{diff} is ~ 2 k Ω in Fig. 3. No negative R_{diff} could be seen in Fig. 4, even when the power of the rf-drive is very low. So, I doubt the treatment of the data.

2. I have serious concerns on the measurement method. I learned from the supplementary that the measurement was performed in a current-driven mode (voltage applied through a large resistor). To measure Shapiro steps, voltage steps with a constant voltage over a range of current, current-driven mode is appropriate since one wants to probe the current range of the voltage plateau. In the opposite, to measure the dual Shapiro steps, current steps with a constant current over a range of voltage, voltage-driven mode is appropriate. It seems that the choice of the measurement method violates the scope. Without the correct measurement method, the conclusions are questionable.

Other issues:

3. The authors showed the rf power at the lowest temperature stage, which is inappropriate. They mentioned that the total attenuation is 50 dB and ignored the losses from cables and connectors. However, they also mentioned that the loss will be enhanced when the frequency increases, as expected. Therefore, the 50 dB attenuation should not be used for all frequencies. The power at the generator should be given.

4. In the caption of Fig. 3, the authors stated that "The amplitude of the Coulomb blockade and the dual Shapiro steps vary with p_{drive} in qualitative agreement with the results of Ref. [30] and with the Tien-Gordon formula [22, 37]." Without plotting and analyzing the data, such statement is abrupt. The Tien-Gordon formula is applicable to tunneling of particles, and shows typical Bessel function dependence (second order). However, for Shapiro steps in a normal Josephson junction, the dependence is to the first order. My confusion is what is the regime of the current work? Cooper pairs function as particles or macroscopic quantum state of superconductors, or intermediate regime? These points should be explained explicitly.

5. The onset current of LZT and how LZT affects the behavior of the device should be explained explicitly.

6. Page 4, “0.93 (1.53) GHz”, GHz should be nA.

7. Page 4, “for synchronization at higher frequencies larger power is needed [cf. Fig. 3(c)].” Fig. 3(c) is measured at a fixed frequency, and the statement could not be concluded from this figure.

8. A similar work can be found at <https://doi.org/10.21203/rs.3.rs-3848621/v1>.

Reviewer #2 (Remarks to the Author):

Dear authors,

In the manuscript entitled “Demonstration of dual Shapiro steps in small Josephson junction”, the authors fabricate a device in which it is possible to observe discrete current steps in the current-voltage characteristics when an additional ac drive frequency is applied to an existing Josephson junction – the so-called dual Shapiro steps. Their experiments reveal dual Shapiro steps up to drive frequencies of 6 GHz. A clever design, building on their earlier work (<https://doi.org/10.1103/PhysRevLett.132.027001>), allows for this visualization by placing a standard Al/AlOx/Al Josephson junction in a high impedance environment comprised of TiOx and granular Al leads. Furthermore, the authors experimentally show that it is possible to modulate the response of the junction, more precisely the height of the differential resistance peak, by changing the input high frequency signal.

Overall, the presented manuscript contains a detailed investigation of the underlying Physics and device requirements needed for the visualization of the dual Shapiro steps, which are convincingly observed. However, it is important to stress here that this is not the first demonstration of the phenomenon or, in other words, the quantum duality effects in Josephson junctions had already been previously confirmed. The authors themselves cite references [14] and [22], but the reader may be led to think otherwise by affirmations in the abstract and conclusions. In our opinion, the main novelty of the presented manuscript apparently lies in the pulsed drive excitation, which potentially allows for an improved and tailored Shapiro response, but this part of the manuscript lacks clarity. We would also like to call the author’s attention to a competing work we came across while analyzing the submitted manuscript: <https://www.researchsquare.com/article/rs-3848621/v1> (The quantised current steps due to the synchronisation of microwaves with the Bloch oscillations in small Josephson junctions, by Antonov et al.). Although the authors were probably unaware of this work during their experiments, we feel obliged to mention that the above manuscript presents analogous results with frequencies up to 24.

Considering the overall picture and the fact that the synchronization of Bloch oscillations has already been experimentally realized, we believe the present manuscript lacks the extremely high novelty standards desired for publication in Nature Communications and therefore do not recommend its publication at this time. This is not a comment on the quality of the work, which we believe presents unambiguous evidence and solid interpretation of the studied phenomena, and, as such, will certainly be beneficial for researchers working on quantum metrology.

Regarding the presentation of the work, we would like to share a few comments we hope will help in

enhancing the presentation of the results.

The theory presented could be more well explored in such a way to enhance the results and guide the design parameters of future devices. Specifically, it would be beneficial to directly link the model to experiments thus trying to understand the impact of different parameters of the rf excitation and device design in the synchronization of the Bloch oscillations and, consequently, on the dual Shapiro steps. In a sense, the authors already show it is possible to tune the dynamics of the system by their pulsed drive experiments, but is it possible to predict these responses to find optimal operation conditions? The authors show results of modelling in the supplement material therefore it is possible to use this tool to conduct a study on these operation conditions. We believe the main text would benefit from these discussions.

In this spirit, one interesting aspect of the author's design is the presence of the SQUID, which allows for additional tunability of the system. It would be interesting to see a systematic investigation of the effect of the SQUID showing the parameters selected are indeed optimal to observe oscillations.

Lastly, although this is a very interesting topic, we believe the authors can enhance their discussion of the pulsed drive experiments. In the discussion, the authors introduce these experiments stating they were performed "for a more specific study", but what is the added value of this study? How is the pulsed drive regime "dual to the single flux quantum mode of Josephson oscillations"? This does not become clear by referring to references [28-30]. Moreover, on top of enhancing the R_{diff} peak, how the modulation of the Shapiro response by different drives can be explored in quantum metrology? As a suggestion, we also believe the text (Supplemental) would benefit from a figure directly comparing the widths of the Shapiro steps in the case of sinusoidal and pulsed excitation, akin those in [28]. This is particularly the case because the steps are not sharp in the I-V curves presented in Fig. 3 (for instance in comparison to those in Ref. [22]).

Yours sincerely,

Reviewer #3 (Remarks to the Author):

Paper's summary, key results and intended audience:

The paper presents an experiment where dual Shapiro steps are observed in a SQUID inserted in a high impedance environment. This impedance is provided by a combination of high kinetic inductance made out of granular aluminium (GrAl) and resistors made out of TiOx. In addition, they show that pulse shaping can modify the step profile, analogous to what is observed with Shapiro step. They also argue that this pulse shaping technique could be useful for metrology.

The paper is very well written and easy to follow. The data analysis together with the interpretations are sound. The experiment is well explained for the experiment to be replicated.

The choice of materials and pulse shaping are the main novelties of this article. This article may be relevant both to superconducting circuits and to the metrology community. I think the first point is interesting in the sense that the paper confirms observations made in two previous papers on a slightly

different platform [1, 2]. As far as the pulse shaping technique is concerned, it might be interesting to develop it a little further to strengthen the paper (see my questions/comments).

Related to the editor question about how does this experiment compare to [3]. While the materials are similar the circuit layout is different. It would be interesting to elaborate on the reason why they changed it to use a more conventional technique. Or if they are planning on using this technique to improve the current setup.

I will leave it to the editors to decide whether these novelties conform to Nature Communication standards.

I think the quality of the article could easily be strengthened by taking into account the comments below.

Major comments/Questions:

- It is not explained how the inductance of the GrAl (and its corresponding characteristic impedance Z_{GrAl}) and the resistance of the TiOx are extracted. As these are the most important design parameters for this experiment, I recommend the authors to be more specific about how they are estimated.

- Since the inductance is fairly small compared to the two previous papers reporting dual Shapiro steps[1,2] it could be interesting to discuss the reasoning behind the choice of L_{GrAl} (and consequently C_{GrAl}) and R_{TiOx} . I guess in theory you want to make R_{TiOx} as small as possible compared to Z_{GrAl} for the dissipation to be as small as possible (and hence the steps to be as sharp as possible) without suffering from too much from the resulting shunting capacitance?

- It is argued that since the fridge base temperature is at 15mK, $kT \ll \min(E1) - \max(E0)$ but is it true that the TiOx resistance is thermalized at 15mK. In particular, the temperature of the latter could be higher due to Joule heat caused by the bias. Has this been evaluated?

- Linked to this question it would be nice to have a more quantitative discussion on how the heating broaden the step. i.e. given the design parameters can we estimate how much of the broadening is due to the heating effect. Furthermore, what choice of design parameter would mitigate this effect?

- Since you measure $I(f)$ for continuous range of frequency it could be nice to express the extracted $2e$ value and its uncertainty to compare its accuracy with respect to the previous study[1,2].

- Linked to that it would improve the paper quality if you could discuss in more detail the accuracy needed for a metrological application, compared to the one you have with the current setup. Together with ways you can think off to increase this accuracy.

- Regarding the pulse shaping technique it is not clear what the authors intend to prove. If is to show that the duality extend to this level it is fine. However they also write that "The higher $R_{\text{diff, peak}}$ values indicate the improved flatness of the dual Shapiro steps." in Fig.4, which I think is questionable. The higher $R_{\text{diff, peak}}$ value can also show a bigger step if not correlated with a sharper peak. Hence,

a more convincing way to display it would be to show the result of the fit using a gaussian function with and without this technique. Another way to show if this new scheme has potential for metrology could be to extract the $2e$ and its uncertainty with and without this new scheme and show that the value is indeed more accurate. If it turns out not to be the case may be this new scheme needs to be refined.

Minor comments:

- In Fig.1 it would be helpful to have an equivalent circuit of the chip, i.e what is the layout of the different inductances, resistors and capacitance around the SQUID with their associated values.

- Since E_j and E_c of the SQUID are estimated from the I-V curve at zero and half flux it would be interesting to check that the estimated makes sense with the measured?

- It is claimed that the GrAI act as quasiparticle filter. While this is true, the authors do not explain how this is relevant to this experiment.

- As shown in Fig 1.a and b the authors choose to be in the $E_c/E_j \sim 2.3$ regime where the Bloch bands are not pure cosine. May be it would be worth mentioning if it can affect the Bloch oscillation dynamics.

- There is a typo in the main text " $I = \pm 2efdrive \approx 0.93$ (1.53) GHz" should be replaced by " $I = \pm 2efdrive \approx 0.93$ (1.53) nA"

- Linked to this: I do not understand the convention the authors follow for the uncertainty since none of them seems to follow on the standard form, but I have always been confused with this.

(https://www.bipm.org/documents/20126/2071204/JCGM_100_2008_E.pdf/cb0ef43f-baa5-11cf-3f85-4dcd86f77bd6#page=38).

[1] - R. S. Shaikhaidarov, K. H. Kim, J. W. Dunstan, I. V. Antonov, S. Linzen, M. Ziegler, D. S. Golubev, V. N. Antonov, E. V. Il'ichev, and O. V. Astafiev, Quantized current steps due to the ac coherent quantum phase-slip effect, *Nature* 608, 45 (2022)

[2] - N. Crescini, S. Cailleaux, W. Guichard, C. Naud, O. Buisson, K. W. Murch, and N. Roch, Evidence of dual shapiro steps in a josephson junction array, *Nature Physics* , 1 (2023).

[3] - F. Kaap, D. Scheer, F. Hassler, and S. Lotkhov, On chip synchronization of bloch oscillations in a strongly coupled pair of small josephson junctions (2023), arXiv:2306.06996 [cond-mat.mes-hall].

Dear reviewers,

We would like to take the opportunity to thank you for reviewing our manuscript. We appreciate the effort of assessing the manuscript and giving important suggestions how it could be improved.

In the following we give a point-by-point reply to your points of critique and believe that our changes in the manuscript, which are in part supported by additional measurements, improve its clarity and strengthen its overall message.

Best regards,

On behalf of all authors,

Fabian Kaap, Lukas Grünhaupt, Sergey Lotkhov

Reviewer #1

In this work, the authors studied the dual Shapiro steps stemming from the synchronization of Bloch oscillations and an external rf-drive. The experiment was performed in Josephson junctions made of Al/AlOx/Al, embedded in a high impedance environment. Quantized, but with a finite slope, current steps were observed. They further investigated the effect of pulsed drives of different forms, and reached a consistent behaviour, especially the asymmetric pattern. However, at the current stage, I could not make a decision on the recommendation of publication since I have some serious concerns.

Response: We appreciate the reviewer's assessment that we reached consistent behaviour in our experimental work especially with regards to the pulsed drive. In our further replies to the reviewer's comments, we believe to have addressed all previous concerns warranting a recommendation to publish our manuscript in its revised form.

1. I have serious concerns on the data. Without the rf-drive, there is back-bending in the I-V curve, e.g., in Fig. 2. When the rf-drive is applied, the back-bending is reformed and the current steps appear, e.g., in Fig. 3. However, the back-bending does not disappear completely and is still there. Therefore, the differential resistance should have negative values. However, the minimum R_{diff} is ~ 2 kOhm in Fig. 3. No negative R_{diff} could be seen in Fig. 4, even when the power of the rf-drive is very low. So, I doubt the treatment of the data.

Response: The reviewer is completely right with their comment, and we thank them for their detailed consistency check of our figures. In the previous version of the manuscript, we had plotted the amplitude of the lock-in response instead of the in-phase amplitude. Not showing the phase of the lock-in amplitude, which indeed shows a 180-degree phase flip in the region of negative differential resistance led to the apparent inconsistency between the IV-curve and the differential resistance plots. We want to stress, however, that no further processing of the lock-in data is performed.

Following the reviewer's comment, we adjusted all plots of lock-in amplifier data and now show the in-phase amplitude. In our updated plots the negative resistance coincides with

the IV-curve of Fig. 2(a) as expected. In addition, thanks to the reviewers comment the peaks in differential resistance have now improved visibility. To further address the reviewer's concern regarding the data we have uploaded all data used for the plots as .txt files to a repository. In addition, we have also uploaded the databases of our experiments comprising all measurements to a public repository, see [10.5281/zenodo.11919736](https://doi.org/10.5281/zenodo.11919736).

2. I have serious concerns on the measurement method. I learned from the supplementary that the measurement was performed in a current-driven mode (voltage applied through a large resistor). To measure Shapiro steps, voltage steps with a constant voltage over a range of current, current-driven mode is appropriate since one wants to probe the current range of the voltage plateau. In the opposite, to measure the dual Shapiro steps, current steps with a constant current over a range of voltage, voltage-driven mode is appropriate. It seems that the choice of the measurement method violates the scope. Without the correct measurement method, the conclusions are questionable.

Response: We want to note that for any DC-transport measurement with a galvanic connection between a voltage source and device under test a current will flow depending on the total resistance of the measurement circuit. In this sense it is unclear to us how the reviewer envisions to implement a purely voltage-bias mode. Furthermore, we do use a voltage source to bias our circuit. However, since Bloch oscillations require a high-impedance environment of leads, applying a voltage will invariably lead to a current flowing through the sample as described above. The additional resistors at room temperature are used to provide a bias current in the appropriate range, which would otherwise approach the resolution limit of our voltage source. Furthermore, any negative resistance branch might lead to hysteretic behaviour in the VI-curve and ambiguous currents for certain bias voltages. In addition to these fundamental arguments, we want to point out to the reviewer, that previous work demonstrating dual Shapiro steps also used a current bias scheme [8,9,22]. At the very least in these works the voltage was applied solely through the on-chip resistors, i.e. potentially employing a different total resistance to that used in our experiment, but fundamentally the same measurement methodology.

3. The authors showed the rf power at the lowest temperature stage, which is inappropriate. They mentioned that the total attenuation is 50 dB and ignored the losses from cables and connectors. However, they also mentioned that the loss will be enhanced when the frequency increases, as expected. Therefore, the 50 dB attenuation should not be used for all frequencies. The power at the generator should be given.

Response: For different measurements it was necessary to vary the attenuation inside the cryostat, such that the power levels at the output of the rf-generator or AWG do not tell the full story. For experimentally reproducing our results we believe it is more instructive to provide an estimate of the power reaching the sample box. Of course, the reviewer is right that the attenuation provided by the cabling is neglected. However, we assume that the power losses due to the cabling are rather small (~3 dB). The power we show is thus the power, that reaches the RF-input of the sample box. To clarify this, we expanded the phrase in the manuscript: "The rf-powers quoted in the following refer to

the power level at the mK-stage, where additional cable losses are neglected and should be equal for all experiments.”

4. In the caption of Fig. 3, the authors stated that “The amplitude of the Coulomb blockade and the dual Shapiro steps vary with p_{drive} in qualitative agreement with the results of Ref. [30] and with the Tien-Gordon formula [22, 37].” Without plotting and analyzing the data, such statement is abrupt. The Tien-Gordon formula is applicable to tunneling of particles, and shows typical Bessel function dependence (second order). However, for Shapiro steps in a normal Josephson junction, the dependence is to the first order. My confusion is what is the regime of the current work? Cooper pairs function as particles or macroscopic quantum state of superconductors, or intermediate regime? These points should be explained explicitly.

Response: We agree with the reviewer that mentioning of the Bessel behaviour in the caption was abrupt and thus moved the discussion to the main text. The reviewer raises an excellent question on the exponent of the Bessel function, which is still a subject of theoretical discussion to our knowledge. Therefore, we added a reference to a recent arXiv-preprint (Ref. [40] in the updated manuscript), which discusses this exponent of the Bessel-like behaviour. Following the theory outlined in this preprint a meaningful investigation of the exact behaviour including the order of the Bessel function requires an environment of tuneable impedance. This is outside the scope of our current work and should be the subject of further studies with dedicated sample design to elucidate such details. Additionally, since our implementation is limited by heating when applying the rf-drive we have access to a limited power range, which would not allow us to extract an exact value of the exponent, which is why we refrain from doing it.

5. The onset current of LZT and how LZT affects the behavior of the device should be explained explicitly.

Response: We thank the reviewer for bringing this omission to our attention and have added a sentence to the third paragraph of page three to clarify this point. The additional text reads: ‘The LZT manifests in a resistive branch in the IV -characteristics due to the excitation of the system to higher energy bands leading to a breakdown of the ground-state approximation necessary for Bloch oscillations.’

6. Page 4, “0.93 (1.53) GHz”, GHz should be nA.

Response: We thank the reviewer for pointing out this error and corrected it in the manuscript.

7. Page 4, “for synchronization at higher frequencies larger power is needed [cf. Fig. 3(c)].” Fig. 3(c) is measured at a fixed frequency, and the statement could not be concluded from this figure.

Response: Indeed, the statement that higher power is necessary for larger frequencies cannot be directly concluded from Fig. 3(c). What we meant to convey by referring to Fig. 3(c) was that the appearance of the different orders of dual Shapiro steps is governed by the Bessel-like behaviour previously mentioned. Theoretically the rf-drive amplitude

enters the argument of the Bessel function, where it is normalized to the drive frequency. Thus, increasing the drive frequency at a constant power reduces the effective strength of the first dual Shapiro step in Fig. 3(b). To make this more explicit we added a discussion in the main text, covering the predicted power dependence of the dual Shapiro steps. The added text reads: ‘Additionally, for synchronization at higher frequencies larger power is needed, since the width of (dual) Shapiro steps scales proportional to $|J_n(\alpha)|^b$, where J_n are the Bessel functions of the first kind, $\alpha \propto (P_{\text{drive}})^{1/2}/f_{\text{drive}}$ is a normalized power and b is an exponent, which depends on the impedance of the environment [22, 40].’

8. A similar work can be found at <https://doi.org/10.21203/rs.3.rs-3848621/v1>.

Response: We thank the reviewer for bringing this work to our attention, which we had previously missed. We agree that the manuscript by Antonov et al. shares similarity with our manuscript. However, it does not investigate different rf-driving approaches, nor does it show appearance of dual Shapiro steps over a continuous frequency range of 5 GHz. It appears to us, in addition, that our results contradict the claim of their preprint that no microwave response can be observed for a blockade voltage above 5 μV . In the interest of scientific transparency we added a reference to this work after the conclusion of our revised manuscript, which reads: ‘During the review of our manuscript we became aware of a preprint by Antonov et al. [42], demonstrating dual Shapiro steps in Al/AlOx/Al Josephson junctions. However, we note that our result seem to contradict the observation of Antonov et. al that dual Shapiro steps are only observed in samples with a Coulomb blockade smaller than 5 μV .’

Reviewer #2 (Remarks to the Author):

In the manuscript entitled “Demonstration of dual Shapiro steps in small Josephson junction”, the authors fabricate a device in which it is possible to observe discrete current steps in the current-voltage characteristics when an additional ac drive frequency is applied to an existing Josephson junction – the so-called dual Shapiro steps. Their experiments reveal dual Shapiro steps up to drive frequencies of 6 GHz. A clever design, building on their earlier work (<https://doi.org/10.1103/PhysRevLett.132.027001>), allows for this visualization by placing a standard Al/AlOx/Al Josephson junction in a high impedance environment comprised of TiOx and granular Al leads. Furthermore, the authors experimentally show that it is possible to modulate the response of the junction, more precisely the height of the differential resistance peak, by changing the input high frequency signal.

Overall, the presented manuscript contains a detailed investigation of the underlying Physics and device requirements needed for the visualization of the dual Shapiro steps, which are convincingly observed.

Response: We thank the reviewer for their summary of our manuscript. Especially, we appreciate that the reviewer evaluates very positively our investigation of underlying physics and device requirements and calls our observation of dual Shapiro steps convincing. Their comment reinforces our conviction that indeed Nature Communications is the appropriate venue for the publication of our manuscript.

However, it is important to stress here that this is not the first demonstration of the phenomenon or, in other words, the quantum duality effects in Josephson junctions had already been previously confirmed. The authors themselves cite references [14] and [22], but the reader may be led to think otherwise by affirmations in the abstract and conclusions.

Response: We agree with the reviewer that indeed quantum duality effects in Josephson junctions had previously been confirmed. As early as 1991 or 1994, respectively, the first results were published (Refs. [8,9]). These early results were, in our opinion, limited by severe heating and as a result limited clarity. Regardless, we believe to do this groundbreaking results justice by mentioning them in the second paragraph of the manuscript. Regarding the recent demonstrations of Refs. [14, 22], our intention was not to lead the reader to believe that our results appeared before the respective publications. We have rephrased the abstract and third paragraph of the revised manuscript in an effort to state this with more clarity, while pointing out the key differences of these results to our work.

In our opinion, the main novelty of the presented manuscript apparently lies in the pulsed drive excitation, which potentially allows for an improved and tailored Shapiro response, but this part of the manuscript lacks clarity.

Response: We thank the reviewer for appreciating the novelty of the pulsed drive excitation, which we not only experimentally investigated, but also numerically modelled. The demonstration and modelling reinforce the duality with Shapiro step

experiments, as well as leads to more pronounced dual Shapiro steps. As such we believe this fact already presents a significant novelty. Regarding the perceived lack of clarity of the tailored Shapiro response description we believe the revised manuscript has several improvements over the previous version and now presents this important part of our work adequately.

We would also like to call the author's attention to a competing work we came across while analyzing the submitted manuscript: <https://www.researchsquare.com/article/rs-3848621/v1> (The quantised current steps due to the synchronisation of microwaves with the Bloch oscillations in small Josephson junctions, by Antonov et al.). Although the authors were probably unaware of this work during their experiments, we feel obliged to mention that the above manuscript presents analogous results with frequencies up to 24.

Response: We thank the reviewer for bringing this work to our attention. As already mentioned in our reply to reviewer #1, indeed the work by Antonov et al. shares similarity with our manuscript. However, it does not investigate different rf-driving approaches, nor does it show appearance of dual Shapiro steps over a continuous frequency range of 5 GHz. It appears to us, in addition, that our results contradict the claim of their preprint that no microwave response can be observed for a blockade voltage above 5 μ V. Taking a closer look at the preprint by Antonov et al. and its associated supplementary, we fail to find evidence for their claim of frequencies up to 24 GHz besides a mention in the text. The highest rf-drive frequency for which data is presented is 10.215 GHz, with only one other plot shown for a rf-drive of 6.495 GHz. In the interest of scientific transparency we added a reference to this work after the conclusion of our revised manuscript, which reads: 'During the review of our manuscript we became aware of a preprint by Antonov et al. [42], demonstrating dual Shapiro steps in Al/AlOx/Al Josephson junctions. However, we note that our result seem to contradict the observation of Antonov et. al that dual Shapiro steps are only observed in samples with a Coulomb blockade smaller than 5 μ V.'

Considering the overall picture and the fact that the synchronization of Bloch oscillations has already been experimentally realized, we believe the present manuscript lacks the extremely high novelty standards desired for publication in Nature Communications and therefore do not recommend its publication at this time. This is not a comment on the quality of the work, which we believe presents unambiguous evidence and solid interpretation of the studied phenomena, and, as such, will certainly be beneficial for researchers working on quantum metrology.

Response: We thank the reviewer for so highly recommending our results and calling them unambiguous evidence and solid interpretation of the studied phenomena. Regarding the lack of novelty, we respectfully disagree with the reviewer's assessment. Recently the field of dual Shapiro steps was reawakened by publications in Nature (Shaikhaidarov et al., Ref [22]) and Nature Physics (Crescini et al., Ref [14]), both journals which require even higher novelty standards, despite the fact that indication of synchronization of Bloch oscillations was experimentally realized as early as 1991 (Ref. [8] in our revised manuscript).

Compared to Ref. [22] we demonstrate dual Shapiro steps in an Al/AlOx/Al Josephson junction system, closely following the original proposal of Likharev & Zorin. In addition, Shaikhaidarov et al. had claimed that clear observation of dual Shapiro steps in a tunnel junction system would be ‘prevented by unavoidable broadening’ (second to last phrase in abstract), which is clearly not the case as confirmed by our results. Furthermore, realization of the dual Shapiro step effect in a CQPS junction comes with challenging fabrication as well as limited yield (“Out of a total 32 measured samples, 9 exhibit dual Shapiro steps[...]” (main text); “One should optimise nanofabrication techniques” (supplementary)).

Compared to the results by Crescini et al., it is worth mentioning that due to their implementation of the high impedance environment they were limited to rf-drives at four distinct frequencies between 3.2 GHz and 6.4 GHz. We not only show that a different implementation of the high impedance environment and delivery of the rf-drive massively extends the range, where dual Shapiro steps in Al/AlOx/Al Josephson junctions can be demonstrated, but also that a pulsed drive realizes an asymmetric pattern of dual Shapiro steps with increased differential resistance peak height. Furthermore, we believe it is of high importance for the scientific community to publish works confirming, and – as is the case for our current manuscript – going beyond what had previously been demonstrated.

Considering all the above factors we remain convinced, that our work constitutes a significant novelty for the field and that Nature Communications is the appropriate journal.

The theory presented could be more well explored in such a way to enhance the results and guide the design parameters of future devices. Specifically, it would be beneficial to directly link the model to experiments thus trying to understand the impact of different parameters of the rf excitation and device design in the synchronization of the Bloch oscillations and, consequently, on the dual Shapiro steps. In a sense, the authors already show it is possible to tune the dynamics of the system by their pulsed drive experiments, but is it possible to predict these responses to find optimal operation conditions? The authors show results of modelling in the supplement material therefore it is possible to use this tool to conduct a study on these operation conditions. We believe the main text would benefit from these discussions.

Response: Indeed, as the reviewer notices we show that it is possible to tune the dynamics of the system by tailoring the drive. Furthermore, we have used a minimal model to numerically predict the response of the system to pulsed driving. With regards to a study on the operation conditions we have now performed additional measurements to extract the electron temperature of the device. Without an rf-drive we estimate an electron temperature of 40 mK. However, when applying the rf-drive, the electron temperature is estimated to be around 200 to 250 mK. The modelling of the expected dual Shapiro steps done in the supplementary is assuming $T=0$. Therefore, before optimizing further the specific design parameters such as R , L , E_C , E_J , or the drive shape we need to improve the heating issue during rf-driving. Solving this will then enable us to do what the reviewer is requesting, namely a detailed study how to optimize the circuit layout for optimal operation conditions. Still, our model is already able to qualitatively explain the enhancement and asymmetric behaviour of the dual Shapiro steps during pulsed driving.

In this spirit, one interesting aspect of the author's design is the presence of the SQUID, which allows for additional tunability of the system. It would be interesting to see a systematic investigation of the effect of the SQUID showing the parameters selected are indeed optimal to observe oscillations.

Response: We thank the reviewer for this suggestion and consequently performed additional measurements of the steps at fixed frequency and power while varying external magnetic field. We see that close to $\Phi_e/\Phi_0 \approx 0$ and thus high Josephson energies the steps are small and measurement noise prevents a clear observation of the steps. In the intermediate regime ($\Phi_e/\Phi_0 \approx 0$, around 6.5 mA bias current through the coil) the steps are clearly visible and Landau-Zener tunnelling has a small influence on the steps. At higher currents the steps are larger but start to overlap with the LZT-branch. We added a plot to the supplementary (Fig. 7), which explains our choice of the coil current.

For additional transparency we have published all experimental data on a public repository ([10.5281/zenodo.11919736](https://zenodo.org/record/11919736)). This also comprises measurements of 17 different flux bias points, each measuring from 1 to 6 GHz at fixed power.

Lastly, although this is a very interesting topic, we believe the authors can enhance their discussion of the pulsed drive experiments. In the discussion, the authors introduce these experiments stating they were performed “for a more specific study”, but what is the added value of this study? How is the pulsed drive regime “dual to the single flux quantum mode of Josephson oscillations”? This does not become clear by referring to references [28-30]. Moreover, on top of enhancing the R_{diff} peak, how the modulation of the Shapiro response by different drives can be explored in quantum metrology? As a suggestion, we also believe the text (Supplemental) would benefit from a figure directly comparing the widths of the Shapiro steps in the case of sinusoidal and pulsed excitation, akin those in [28]. This is particularly the case because the steps are not sharp in the I-V curves presented in Fig. 3 (for instance in comparison to those in Ref. [22]).

Response: To clarify what is meant by “dual to the single flux quantum mode of Josephson oscillations”, we use the analogous to the voltage standard based on Josephson junction. Each period of drive oscillations or in our case each pulse, leads to a transfer of one Cooper pair instead of one flux quantum. This might enable a variety of dualities between Josephson voltage standard applications and applications based on dual Shapiro steps, such as the dual to Josephson arbitrary voltage synthesizers (JAWS). Following the reviewer's advice we now focus the discussion on the increased accuracy provided by the pulsed drive. We added a plot as the inset of Figure 4(f), where we extract the electron charge via the relation $I = 2ef$ to show the improvement by using a pulsed drive in a more obvious way. For the sinusoidal driving we extract $e = (1.612 \pm 0.010) \times 10^{-19}\text{C}$ and for the pulsed driving $e = (1.610 \pm 0.005) \times 10^{-19}\text{C}$.

Note, that the y-errors are the fitting errors of the extracted position of the dual Shapiro steps using a Gaussian fit (Fig. S4), i.e. the errors on the peak position extracted from the pulsed driving dataset are visibly smaller. We have added clarification of this by writing: ‘The extracted value of the electron charge for the pulsed drive $e_{\text{pulse}} = (1.610 \pm 0.005) \times 10^{-19}\text{C}$ is showing smaller errors than for the sinusoidal case $e_{\text{sine}} = (1.612 \pm 0.010) \times 10^{-19}\text{C}$.’

It is important to stress, however, that the errors on e are given by the statistical error of the measurement and don't include systematic errors. Notably, our results and errors are on par with References [14, 22]. Note, that in [22] the range of the current-axis is a factor of 10 larger (± 30 nA) than ours [± 3 nA, Fig. 3(a)] and thus the steps appear steeper.

Reviewer #3 (Remarks to the Author):

The paper presents an experiment where dual Shapiro steps are observed in a SQUID inserted in a high impedance environment. This impedance is provided by a combination of high kinetic inductance made out of granular aluminium (GrAl) and resistors made out of TiOx. In addition, they show that pulse shaping can modify the step profile, analogous to what is observed with Shapiro step. They also argue that this pulse shaping technique could be useful for metrology. The paper is very well written and easy to follow. The data analysis together with the interpretations are sound. The experiment is well explained for the experiment to be replicated.

Response: We thank the reviewer for their summary of our manuscript and their very positive evaluation. Especially, we are delighted that the reviewer judges our experiment to be well explained to facilitate replication.

The choice of materials and pulse shaping are the main novelties of this article. This article may be relevant both to superconducting circuits and to the metrology community. I think the first point is interesting in the sense that the paper confirms observations made in two previous papers on a slightly different platform [1, 2]. As far as the pulse shaping technique is concerned, it might be interesting to develop it a little further to strengthen the paper (see my questions/comments).

Response: We agree with the reviewer's assessment on the novelties of our work and want to explicitly point out that we were able to demonstrate dual Shapiro steps in a frequency range of 1 to 6 GHz compared to four distinct frequencies in the work of Crescini et al. Thanks to the reviewer's constructive criticism we have performed additional experiments and updated the presentation of some of the data leading to an overall stronger manuscript.

Related to the editor question about how does this experiment compare to [3]. While the materials are similar the circuit layout is different. It would be interesting to elaborate on the reason why they changed it to use a more conventional technique. Or if they are planning on using this technique to improve the current setup. I will leave it to the editors to decide whether these novelties conform to Nature Communication standards. I think the quality of the article could easily be strengthened by taking into account the comments below.

Response: Compared to our earlier work, where we investigated the synchronization of Bloch oscillations in two capacitively coupled circuits, in our current manuscript we aimed to implement the original proposal of Likharev & Zorin (1985) to demonstrate dual Shapiro steps. This serves the purpose of validating our material platform and choice of Josephson junction parameters (size, tunnel barrier oxidation strength). Aside from this fundamental achievement to implement the circuit as theoretically proposed, which allowed us to present dual Shapiro steps in a Josephson junction system over an unprecedented frequency range, the pulsed drive adds significant novelty. It reaffirms the duality to Shapiro steps and enables a more precise extraction of dual Shapiro step position ultimately leading to a factor 2 reduced statistical uncertainty for the elementary charge. Based on the presented results, we plan to develop further a design, which will

allow the observation of quantized current steps without an external rf-generator by using a Josephson oscillator to provide the external rf-drive for locking. Ultimately, we aim to demonstrate dual Shapiro steps routing only DC-signals to the chip.

It is not explained how the inductance of the GrAl (and its corresponding characteristic impedance Z_{GrAl}) and the resistance of the TiOx are extracted. As these are the most important design parameters for this experiment, I recommend the authors to be more specific about how they are estimated.

Response: We thank the reviewer for highlighting this important issue. We measure the resistance R of the TiOx at mK temperature through the leads. Since the Al and grAl are superconducting, the measured resistance is that of the resistors, neglecting the resistance of the Thermocoax wiring, which is on the order of 200Ω . The grAl inductance is estimated from room temperature resistance measurements. Using the Matthis-Bardeen-formula [38], linking the room temperature resistance to the kinetic inductance, we estimate the kinetic inductance of the grAl strips.

To clarify how the numbers quoted in the manuscript are extracted, we added a phrase to the paragraph on page 3 of the revised manuscript: ‘From a measurement of the leads with the cryostat at base temperature, we extract the resistance of the TiOx strips $R_{\text{TiOx}} = 275 \text{ k}\Omega$. Using the Matthis-Bardeen formula we estimate $L_{\text{grAl}} = 0.18\hbar R / (k_{\text{B}}T_{\text{c}}) \approx 50 \text{ nH}$ [38], from a room temperature resistance measurement of the grAl strip.’

Since the inductance is fairly small compared to the two previous papers reporting dual Shapiro steps[1,2] it could be interesting to discuss the reasoning behind the choice of L_{GrAl} (and consequently C_{GrAl}) and R_{TiOx} . I guess in theory you want to make R_{TiOx} as small as possible compared to Z_{GrAl} for the dissipation to be as small as possible (and hence the steps to be as sharp as possible) without suffering from too much from the resulting shunting capacitance?

Response: We agree with the reviewer that intuitively lowering the resistance will lower the dissipation. However, as stated in the response to the reviewer’s next comment below, we performed additional measurements and estimated the temperature of the circuit. We observed an electron temperature of 40 mK without rf-drive. In our circuit, the main element to create the large impedance of the environment is the resistor and not the grAl strip. This is a result of previous experiments, where using a larger resistance and a shorter grAl strip led to an enhanced Coulomb blockade. However, we did not sufficiently investigate the parameter regime enabling the observation of dual Shapiro steps to make a well-founded claim.

The electron temperature while applying the rf-drive was estimated to be around 200 to 250 mK. Since the capacitive coupling of the slot line is placed between the two resistors, in an ideal case the current should only be induced through the SQUID and no rf-current should flow through the resistors. Thus, from our perspective it is not certain, that the dissipation of rf power is indeed increased by using a higher resistor. For future designs we will investigate this point more elaborately with the help of detailed numerical and FEM modelling.

It is argued that since the fridge base temperature is at 15mK, $kT \ll \min(E1) - \max(E0)$ but is it true that the TiOx resistance is thermalized at 15mK. In particular, the temperature of the latter could be higher due to Joule heat caused by the bias. Has this been evaluated?

Response: We thank the reviewer for bringing up this point. It motivated us to measure in an additional cooldown the temperature dependence of the Coulomb blockade. We added this measurement to the supplementary, where we estimate the electron temperature for pure dc-measurements and for the rf-driven ones. The electron temperature in an undriven measurement was estimated to be 40 mK, while it rose to 200 to 250 mK during the rf-driving. Even at 250 mK we are still within the limit of $kT = 21 \mu\text{eV} < (\min(E1) - \max(E0)) \sim 36 \mu\text{eV}$.

Linked to this question it would be nice to have a more quantitative discussion on how the heating broaden the step. i.e. given the design parameters can we estimate how much of the broadening is due to the heating effect. Furthermore, what choice of design parameter would mitigate this effect?

Response: The reviewer raises an excellent point, which we now discuss further in the revised manuscript. Based on the estimation of the circuit's temperature with the rf-drive applied, and comparison with the slope of the blockade without rf-drive (see. Fig. 2(b)), we conclude that the broadening is dominated by heating induced by the rf-drive. As mentioned previously, a future task will be to optimize the rf-coupling such that the heating effect due to the rf-drive is minimized. Which design parameter or combination of parameters will lead to achieving this goal is still an open question and will have to be investigated using detailed numerical and finite element modelling.

Since you measure $I(f)$ for continuous range of frequency it could be nice to express the extracted $2e$ value and its uncertainty to compare its accuracy with respect to the previous study[1,2].

Response: We appreciate the reviewer's suggestion and added this plot as the inset of Figure 4(f). For the sinusoidal driving we extract $e = (1.612 \pm 0.010) \times 10^{-19}\text{C}$ and for the pulsed driving $e = (1.610 \pm 0.005) \times 10^{-19}\text{C}$. Note, that the y-errors are the fitting errors of the extracted position of the dual Shapiro steps using a Gaussian fit (Fig. S4), i.e. the errors on the peak position extracted from the pulsed driving dataset are visibly smaller. We have added clarification of this by writing: 'The extracted value of the electron charge for the pulsed drive $e_{\text{pulse}} = (1.610 \pm 0.005) \times 10^{-19}\text{C}$ is showing smaller errors than for the sinusoidal case $e_{\text{sine}} = (1.612 \pm 0.010) \times 10^{-19}\text{C}$.'

The errors are given by the statistical error of the measurement and don't include systematic errors at this point. Reference [1] reported $2e = (3.20 \pm 0.01) \times 10^{-19}\text{C}$ and [2] $2e = (3.23 \pm 0.07) \times 10^{-19}\text{C}$. Therefore, we are on par with previous results.

Linked to that it would improve the paper quality if you could discuss in more detail the accuracy needed for a metrological application, compared to the one you have with the current setup. Together with ways you can think off to increase this accuracy.

Response: We thank the reviewer for highlighting this important point. To make a statement regarding the metrological accuracy needed, we compare to the current state of the art in single electron pumps in the conclusion. For a competitive accuracy an uncertainty of 1 ppm or better should be achieved. We discuss in the conclusion several potential technical improvements to work towards metrological application. To provide more insight regarding this topic, we have added a phrase to the conclusion, which reads: ‘With those improvements, it might be possible to reach the current metrological precision of current standards realized via single electron pumps (≈ 0.2 ppm [6])’

Regarding the pulse shaping technique it is not clear what the authors intend to prove. If is to show that the duality extend to this level it is fine. However they also write that “The higher $R_{\{diff, peak\}}$ values indicate the improved flatness of the dual Shapiro steps.” in Fig.4, which I think is questionable. The higher $R_{\{diff, peak\}}$ value can also show a bigger step if not correlated with a sharper peak. Hence, a more convincing way to display it would be to show the result of the fit using a gaussian function with and without this technique. Another way to show if this new scheme has potential for metrology could be to extract the $2e$ and its uncertainty with and without this new scheme and show that the value is indeed more accurate. If it turns out not to be the case may be this new scheme needs to be refined.

Response: We thank the reviewer for this suggestion. By adding the desired analysis, we verified that the pulse scheme leads to a factor of two smaller error for the extracted electron charge (see caption of Fig. 4(f)). Additionally, the error of the step position extracted from Gaussian fits to the differential resistance measurements is also visibly smaller for the pulsed drive scheme as illustrated by the reduced y-errors in the inset of Fig. 4(f). We now point this out explicitly in the caption of Figure 4: ‘Inset: Measured current of the first dual Shapiro step as a function of the drive frequency for the sinusoidal drive (red and plotted with 0.1 nA offset) and the pulsed drive (beige). From a linear fit we can extract the charge of an electron according to $I = 2efdrive$. For the sinusoidal drive we extract $e = (1.612 \pm 0.010) \times 10^{-19}$ C and for the pulse drive $e = (1.610 \pm 0.005) \times 10^{-19}$ C. Note that the errorbars for the pulsed drive are smaller (see Supplementary Fig. 4).’

In Fig.1 it would be helpful to have an equivalent circuit of the chip, i.e what is the layout of the different inductances, resistors and capacitance around the SQUID with their associated values.

Response: We thank the reviewer for pointing out that an equivalent circuit might help understanding the circuit. We added this to the supplementary (Supplementary Fig. 2) to avoid cluttering Fig. 1(c).

Since E_j and E_c of the SQUID are estimated from the I-V curve at zero and half flux it would be interesting to check that the estimated makes sense with the measured?

Response: We are slightly confused by the reviewer's comment, since it is not clear to us what they mean by estimated and measured. We want to point out, that we performed the measurements shown in the supplementary, which give estimates of E_j and E_c .

It is claimed that the GrAl act as quasiparticle filter. While this is true, the authors do not explain how this is relevant to this experiment.

Response: Since quasiparticle tunnelling could partially replace the coherent Cooper pair transfer, the resulting Coulomb blockade and thus, the dual Shapiro steps would be smaller. We added the following to the text: 'Quasiparticle tunnelling is a process that partially replaces the coherent Cooper pair tunnelling associated with Bloch oscillations. Reducing the presence of quasiparticles in the vicinity of the Josephson junction will thus result in a larger Coulomb blockade.'

As shown in Fig 1.a and b the authors choose to be in the $E_c/E_j \sim 2.3$ regime where the Bloch bands are not pure cosine. May be it would be worth mentioning if it can affect the Bloch oscillation dynamics.

Response: We thank the reviewer for pointing out this topic, since it is indeed a very good question. Since the Bloch oscillations are presumably not perfectly sinusoidal, one could expect differences from a purely cosine potential. By suppressing E_j and thus going to a more sawtooth like potential we could not observe effects, which are not in line with assuming a sinusoidal potential. We think that this is an interesting task for a theoretical analysis. Future experiments with increased precision might reveal such effects.

There is a typo in the main text " $I = \pm 2efdrive \approx 0.93 (1.53) \text{ GHz}$ " should be replaced by " $I = \pm 2efdrive \approx 0.93 (1.53) \text{ nA}$ "

Response: We thank the reviewer for noticing our mistake and have corrected it in the revised manuscript.

Linked to this: I do not understand the convention the authors follow for the uncertainty since none of them seems to follow on the standard form, but I have always been confused with this.

https://www.bipm.org/documents/20126/2071204/JCGM_100_2008_E.pdf/cb0ef43f-baa5-11cf-3f85-4dcd86f77bd6#page=38.

Response: We thank the reviewer for highlighting our misleading notation. We want to point out that by 0.93 (1.53) nA we did not mean to indicate an error but referred to two different currents of the dual Shapiro steps corresponding to two different rf-drive frequencies: 0.93nA and 1.53nA, which are shown in Fig. 3(a). We changed the notation to improve clarity.

I would like to thank the authors for their efforts to address my concerns. Below are my responses. There are still some issues that should be addressed.

1. A serious mistake. Now OK.

2. Let's think of two simplified cases, the Shapiro steps (left figure) and the dual one (right figure). "To measure Shapiro steps, voltage steps with a constant voltage over a range of current, current-driven mode is appropriate since one wants to probe the current range of the voltage plateau. In the opposite, to measure the dual Shapiro steps, current steps with a constant current over a range of voltage, voltage-driven mode is appropriate." We take the left figure as an example: if we sweep current, there are many data points (black squares) that can tell the width of the voltage plateau, and this is what we usually do. Of course, there can be current and voltage whenever we perform the measurement. Do the authors agree with the idea above? If yes, the **starting point** of the experiment should be voltage-driven mode (right figure). If not, a detailed and convincing analysis should be given.

3. "However, they also mentioned that the loss will be enhanced when the frequency increases, as expected." Part of the reply: "we assume that the power losses due to the cabling are rather small (~ 3 dB)". Is this such low? Did the authors measure the losses of the cabling in the range of the frequency used? If this value holds for 1 GHz, does it hold for 6 GHz? (The experimentalists are aware of the sources of the losses, and I believe that the authors should not tell or ignore something that is not sure.)

4. A serious point. "The amplitude of the Coulomb blockade and the dual Shapiro steps vary with p_{drive} in qualitative agreement with the results of Ref. [30] and with the Tien-Gordon formula [22, 37]." I could not find the analysis. It is a drawback of the current work since such analysis could not be accessed.

Overall, I agree with the other reviewers that this work lacks the extremely high novelty standards desired for Nature Communications. I could not recommend its publication.

REVIEWER COMMENTS

Reviewer #1 (Remarks to the Author):

See attached file.

Reviewer #2 (Remarks to the Author):

In their reply to the initial reports, the authors of the manuscript titled “Demonstration of dual Shapiro steps in small Josephson junction” made a strong case for the novelty of the submitted work. Indeed, in our initial report, we praised the work for its scientific accuracy but believed it lacked novelty for a high impact publication in Nature Communications. However, upon considering the authors’ updates to the manuscript and a well-reasoned positioning of their work with respect to the existing literature, we believe that the advancements reported by this manuscript represent a step forward in the comprehension and observations of dual Shapiro steps that could benefit the scientific community, particularly in the field of quantum metrology, with highlight to the results obtained when the device is subjected to the pulsed drive excitation. Moreover, the authors replied satisfactorily to the comments and questions made by the referees and further improved the quality of the presented work by making important new considerations and measurements.

In view of this discussion and the improvements to the manuscript, we believe this work merits publication in Nature Communications.

Reviewer #3 (Remarks to the Author):

The authors have responded to my comments/questions in a detailed and comprehensive manner, and I thank them for that.

In my opinion, the paper can be published in this form.

Reviewer #4 (Remarks to the Author):

Dear reviewers,

We appreciate your time and effort to review the revised version of our manuscript. As stated in our previous reply, we felt that the clarity and strength of our manuscript improved thanks to the feedback by reviewers. We are delighted that Reviewers #2-4 share this assessment of our revised manuscript and now fully support publication in Nature Communications.

In the following we give a point-by-point reply to the remaining points of critique. The citation numbering follows that in the main paper. We hope that these points will be clarified after reading our response.

Best regards,

On behalf of all authors,

Fabian Kaap, Lukas Grünhaupt, Sergey Lotkhov

Reviewer #1

I would like to thank the authors for their efforts to address my concerns. Below are my responses. There are still some issues that should be addressed.

Response: We thank the reviewer for providing additional comments regarding our revised manuscript and giving us the opportunity to clarify further some of the concerns, which were already brought up in the previous round of reviews.

1. A serious mistake. Now OK.

Response: We appreciate that the reviewer is satisfied with the updated figures and our discussion regarding this point in our previous response letter. As discussed there, thanks to the reviewers detailed consistency check of the figures, the peaks in the plots of differential resistance data have improved visibility now.

*2. Let's think of two simplified cases, the Shapiro steps (left figure) and the dual one (right figure). "To measure Shapiro steps, voltage steps with a constant voltage over a range of current, current-driven mode is appropriate since one wants to probe the current range of the voltage plateau. In the opposite, to measure the dual Shapiro steps, current steps with a constant current over a range of voltage, voltage-driven mode is appropriate." We take the left figure as an example: if we sweep current, there are many data points (black squares) that can tell the width of the voltage plateau, and this is what we usually do. Of course, there can be current and voltage whenever we perform the measurement. Do the authors agree with the idea above? If yes, the **starting point** of the experiment should be voltage-driven mode (right figure). If not, a detailed and convincing analysis should be given.*

Response: We thank the reviewer for their effort on further expanding on this point and providing figures to clarify their point further. As a starting point for our experiment, we used the same experimental ansatz as has been done in [8,9,22,42]. In all those and in

our work a bias voltage is applied across a series connection of known high-ohmic resistors and the device under test. The voltage across the sample was either measured directly in a 4-point scheme or calculated by subtracting the voltage drop in the biasing resistor after a 2-point measurement of the sample current. In all the cases, using such biasing mode was found effective in characterizing the peculiarities in the VI-curves related to Bloch oscillations and dual Shapiro steps.

On the contrary, using voltage-bias regime may not be a viable choice for characterizing a complex pattern of dual Shapiro steps. Below we show Figure 3 from the initial publications [4,5], which provides the expected VI-curves. In contrast to the simplified sketch of the reviewer, it can be seen that for the same voltage a large number of current values are possible as discussed in our previous response letter. (Phrase in the previous reply to the reviewers: **Furthermore, any negative resistance branch might lead to hysteretic behaviour in the VI-curve and ambiguous currents for certain bias voltages.**) The ambiguity is however removed, if using a current-bias regime or similar.

Leaving aside the technological aspects outlining that we in-fact use a suitable measurement method and focusing on physics, the starting point is the occurrence of Bloch oscillations in quasi-isolated small Josephson junctions. In the initial work by Averin, Likharev and Zorin [4, 5], they start at a current-bias regime to introduce the coherent Cooper pair transport, namely the Bloch oscillations. Biasing with a current for ramping the junction quasicharge has made the basis of extensive theoretical work done on Bloch oscillations and on the dual Shapiro steps so far [4, 5, 10, 40, ...].

We thus hope to have convinced the reviewer that the used measurement scheme is both suitable and the most appropriate at the present stage of experiments on Bloch oscillations and dual Shapiro steps. In future, more elaborated biasing circuits, for example those including feedback control of the working point (I, V), will be necessary for precise characterization of metrologically flat current plateaus.

3. “However, they also mentioned that the loss will be enhanced when the frequency increases, as expected.” Part of the reply: “we assume that the power losses due to the cabling are rather small (~3 dB)”. Is this such low? Did the authors measure the losses of the cabling in the range of the frequency used? If this value holds for 1 GHz, does it hold for 6 GHz? (The experimentalists are aware of the sources of the losses, and I believe that the authors should not tell or ignore something that is not sure.)

Response: Indeed, we measured the cable losses at room temperature. Below we provide the measured data. Of course, as the reviewer mentions, the attenuation is frequency dependent. In the plot attached below, we show the attenuation of the coax cable going from the electronics to the input of the fridge (Sucoflex, blue color). In orange and green the room temperature transmission of the BeCu-SS cables from the top of the cryostat to the 4K stage are shown. In the cooler temperature environment of the fridge the cable attenuation is even smaller, such that the green and orange curve are upper limits for the cable attenuation. In red the combined curve of the three individual contributions is shown. Therefore, the upper bound for the attenuation at 1 GHz is ~1.7 dB and at 6 GHz ~4.2 dB. The cabling from the 4 K plate to the mixing chamber of the cryostat is via NbTi superconducting coax cables with a loss significantly smaller than 0.01 dB [see <https://doi.org/10.1140/epjqt/s40507-017-0059-7>]. The remaining distance to the sample box input is routed via ~ 30 cm of Cu/Cu coaxial cable, with a loss smaller than 0.17 dB [see <https://doi.org/10.1140/epjqt/s40507-017-0059-7>]. We added the following phrase to the supplementary: Measurements at room temperature yield an upper bound for the frequency dependent losses in the coaxial cables between room temperature electronics and rf input of the chip holder of 1.7dB at 1 GHz and 4.2 dB at 6 GHz. To take the data presented in Fig.4(e) the 10 dB attenuator at the 100mK was removed, since the output voltage of the AWG was limited to 1V.

4. A serious point. “The amplitude of the Coulomb blockade and the dual Shapiro steps vary with p_{drive} in qualitative agreement with the results of Ref. [30] and with the Tien-Gordon formula [22, 37].” I could not find the analysis. It is a drawback of the current work since such analysis could not be accessed.

Response: We respectfully point out to the reviewer that the cited text is not present in the latest version of the manuscript. In our previous reply to the reviewer’s criticism, we addressed this issue in detail and updated the text accordingly. We refer the reviewer to point 4 of our previous reply letter, which we copied below for convenience:

We agree with the reviewer that mentioning of the Bessel behaviour in the caption was abrupt and thus moved the discussion to the main text. The reviewer raises an excellent question on the exponent of the Bessel function, which is still a subject of theoretical discussion to our knowledge. Therefore, we added a reference to a recent arXiv-preprint (Ref. [40] in the updated manuscript), which discusses this exponent of the Bessel-like behaviour. Following the theory outlined in this preprint a meaningful investigation of the exact behaviour including the order of the Bessel function requires an environment of tuneable impedance. This is outside the scope of our current work and should be the subject of further studies with dedicated sample design to elucidate such details. Additionally, since our implementation is limited by heating when applying the rfdrive we have access to a limited power range, which would not allow us to extract an exact value of the exponent, which is why we refrain from doing it.

Overall, I agree with the other reviewers that this work lacks the extremely high novelty standards desired for Nature Communications. I could not recommend publication.

Response: As detailed in our previous response letter (Reply to Reviewer #2 and #3), we present several novel results, which warrant publication of our manuscript in Nature Communications. As can be seen further below, these points convinced Reviewers #2-4, who fully support publication of the manuscript in its current form in Nature Communications.

Reviewer #2

In their reply to the initial reports, the authors of the manuscript titled “Demonstration of dual Shapiro steps in small Josephson junction” made a strong case for the novelty of the submitted work. Indeed, in our initial report, we praised the work for its scientific accuracy but believed it lacked novelty for a high impact publication in Nature Communications. However, upon considering the authors’ updates to the manuscript and a well-reasoned positioning of their work with respect to the existing literature, we believe that the advancements reported by this manuscript represent a step forward in the comprehension and observations of dual Shapiro steps that could benefit the scientific community, particularly in the field of quantum metrology, with highlight to the results obtained when the device is subjected to the pulsed drive excitation. Moreover, the authors replied satisfactorily to the comments and questions made by the referees and further improved the quality of the presented work by making important new considerations and measurements.

In view of this discussion and the improvements to the manuscript, we believe this work merits publication in Nature Communications.

Response: We would like to thank the reviewer for the effort of reviewing our revised manuscript. The question raised by the reviewer were of great help to improve the clarity of our manuscript and presentation of its key findings. We are happy that with these improvements the reviewer acknowledges the novelty of our findings and recommends our work for publication in Nature Communications.

Reviewer #3

The authors have responded to my comments/questions in a detailed and comprehensive manner, and I thank them for that. In my opinion, the paper can be published in this form In view of this discussion and the improvements to the manuscript, we believe this work merits publication in Nature Communications.

Response: We would like to thank the reviewer for taking part in the second round of the reviewing process of our manuscript. Our manuscript profited a lot from their input resulting in a stronger and clearer presentation of our results.

Reviewer #4

Response: We appreciate the time reviewer #4 spent on co-reviewing our manuscript and wish them all the best for their future reviewing activity in particular, and for their scientific career in general.

REVIEWER COMMENTS

Reviewer #1 (Remarks to the Author):

I would like to thank the authors for their efforts to address my concerns. Overall, there are still some issues that the authors could not address. There are drawbacks. This work lacks the extremely high novelty standards desired for Nature Communications. I could not recommend its publication.

The authors wrote “The amplitude of the Coulomb blockade and the dual Shapiro steps vary with p -drive in qualitative agreement with the results of Ref. [30] and with the Tien-Gordon formula [22, 37]” in the first version. So, I believe that they checked this point. However, they removed such statement and they did not explain how their data behave and why in detail. Again, it is a drawback of the current work since such analysis could not be accessed. The dual Shapiro steps are a manifestation, and the underlying physics is that phase and charge are quantum conjugate variables. I wrote “The Tien-Gordon formula is applicable to tunneling of particles, and shows typical Bessel function dependence (second order). However, for Shapiro steps in a normal Josephson junction, the dependence is to the first order. My confusion is what is the regime of the current work? Cooper pairs function as particles or macroscopic quantum state of superconductors, or intermediate regime? These points should be explained explicitly.” They added a new reference regarding the dependence on the microwave. Such analysis gives us a possible tool to diagnose the regime of Cooper pairs, i.e., particles or macroscopic quantum state of superconductors, or intermediate regime. I believe “phase and charge are quantum conjugate variables” is more fundamental.

Dear reviewer,

we appreciate the explanation of your last remaining comment regarding our manuscript. With regards to your previously raised serious concerns on the data and measurement method, we are happy to note that these have been fully resolved now and are no longer of concern.

In the following we present our standpoint on the analysis of the power dependence and your question whether Cooper pairs function as particles or macroscopic quantum states of superconductors, or an intermediate regime.

We hope to have sufficiently clarified this last remaining comment and would like to thank you once more for your time and effort during the review process.

Best regards,

On behalf of all authors,

Fabian Kaap, Lukas Grünhaupt, Sergey Lotkhov

Reviewer #1

The authors wrote “The amplitude of the Coulomb blockade and the dual Shapiro steps vary with p_{drive} in qualitative agreement with the results of Ref. [30] and with the Tien-Gordon formula [22, 37]” in the first version. So, I believe that they checked this point. However, they removed such statement and they did not explain how their data behave and why in detail. Again, it is a drawback of the current work since such analysis could not be accessed. The dual Shapiro steps are a manifestation, and the underlying physics is that phase and charge are quantum conjugate variables. I wrote “The Tien-Gordon formula is applicable to tunneling of particles, and shows typical Bessel function dependence (second order). However, for Shapiro steps in a normal Josephson junction, the dependence is to the first order. My confusion is what is the regime of the current work? Cooper pairs function as particles or macroscopic quantum state of superconductors, or intermediate regime? These points should be explained explicitly.” They added a new reference regarding the dependence on the microwave. Such analysis gives us a possible tool to diagnose the regime of Cooper pairs, i.e., particles or macroscopic quantum state of superconductors, or intermediate regime. I believe “phase and charge are quantum conjugate variables” is more fundamental.

Response: We would like to first clarify what we meant by our previous statement of ‘qualitative (!) agreement’: as can be seen in Fig. 3(c) with increasing drive power the differential resistance peak of the Coulomb blockade decreases, while the peaks of the dual Shapiro steps become visible one after another. This pattern resembles the pattern presented, e.g. in Fig. 4(c) of Ref. [22], but with smaller ranges of data. However, this is obvious to experts in the field. The discussion presented in the first version of the manuscript, which the reviewer’s comments refer to, was not detailed enough to be meaningful for a broader audience. This is why we agreed with the reviewer in our first response letter, that ‘mentioning the Bessel behaviour [...] was abrupt and thus moved

the discussion to the main text.' As outlined in the previous two replies to the reviewer the updated manuscript has a more detailed discussion regarding the power dependence.

Due to the previously discussed heating from the rf-drive, which leads to a suppression of the Coulomb blockade (c.f. the supplementary material Fig. 6) our current data does not allow an accurate extraction of the value for the Bessel function exponent. Therefore, a conclusive interpretation of the R_{diff} vs. $I_{\text{ac}}(P_{\text{drive}})$ dependency in the way it was done by the authors of Ref. [22] using Fig. 4(c) is not justified. Consequently, there is no solid basis to distinguish between the transport mechanisms mentioned by the reviewer and we removed any definitive claims on the exponent which we cannot support quantitatively. We highlighted this by the following phrase in the manuscript: **However, our data does not allow to extract the exact exponent b of this dependence.**

Regarding the more fundamental discussion raised by the reviewer: based on the IV-curves with and without rf-signal [see Figs. 2 and 3(a)] we assume that the oscillatory process realizes an intermediate regime in the following sense. In Fig. 3(a) the two IV-curves showing dual Shapiro steps exhibit pronounced back-bending down from the Coulomb blockade threshold, which is an indication of the (quasi)-coherent ground state transport. However, in the same curves a clear footprint of Landau-Zener tunnelling can be recognised (as discussed in the paper), which involves incoherent Cooper pair relaxation events contributing to the net current. Still, as mentioned above there is no solid basis to claim a definitive regime, which is why we refrain from doing it.

Finally, we stress that the main claim of our manuscript is focussed on the pulsed driving. Our observations in this regime are explained well by using an RLJ-like model dual to that described for the Josephson case in Refs. [29,30]. This basic fact makes us confident about the interpretation of the measured curves in terms of phase-(quasi)charge duality and about the significance of the coherent transport component. Nevertheless, a reliable quantitative analysis of the Bessel function behaviour should provide an important further insight and will be a subject of future experiments with a dedicated sample design, i.e. ideally including an environment of tunable impedance. This hopefully enables to indeed use the theory outlined in the new reference [40] as an additional tool to diagnose the regime of Cooper pairs as suggested by the reviewer.

REVIEWERS' COMMENTS

Reviewer #1 (Remarks to the Author):

I would like to thank the authors again for their efforts to address my concerns. I also checked all the review files again. Overall, I am still a bit hesitated to strongly recommend this paper due to the extremely high novelty standards desired for Nature Communications. At this stage, I would say I am 60% positive on this paper.